# Comparative Transcriptomics in Atlantic Salmon Head Kidney and SHK-1 Cell Line Exposed to the Sea Louse Cr-Cathepsin

**DOI:** 10.3390/genes14040905

**Published:** 2023-04-13

**Authors:** Yeny Leal, Valentina Valenzuela-Muñoz, Antonio Casuso, Bárbara P. Benavente, Cristian Gallardo-Escárate

**Affiliations:** 1Interdisciplinary Center for Aquaculture Research (INCAR), Universidad de Concepción, P.O. Box 160-C, Concepción 4030000, Chile; 2Laboratory of Biotechnology and Aquatic Genomics, Department of Oceanography, Universidad de Concepción, Concepción 4030000, Chile

**Keywords:** *C. rogercresseyi*, recombinant protein, transcriptome response, immune response

## Abstract

The development of vaccines against sea lice in salmon farming is complex, expensive, and takes several years for commercial availability. Recently, transcriptome studies in sea louse have provided valuable information for identifying relevant molecules with potential use for fish vaccines. However, the bottleneck is the in vivo testing of recombinant protein candidates, the dosage, and the polyvalent formulation strategies. This study explored a cell-based approach to prospect antigens as candidate vaccines against sea lice by comparison with immunized fish. Herein, SHK-1 cells and Atlantic salmon head kidney tissue were exposed to the antigen cathepsin identified from the sea louse *Caligus rogercresseyi*. The cathepsin protein was cloned and recombinantly expressed in *Escherichia coli*, and then SHK-1 cell lines were stimulated with 100 ng/mL cathepsin recombinant for 24 h. In addition, Atlantic salmons were vaccinated with 30 ug/mL recombinant protein, and head kidney samples were then collected 30 days post-immunization. SHK-1 cells and salmon head kidney exposed to cathepsin were analyzed by Illumina RNA sequencing. The statistical comparisons showed differences in the transcriptomic profiles between SHK-1 cells and the salmon head kidney. However, 24.15% of the differentially expressed genes were shared. Moreover, putative gene regulation through lncRNAs revealed tissue-specific transcription patterns. The top 50 up and downregulated lncRNAs were highly correlated with genes involved in immune response, iron homeostasis, pro-inflammatory cytokines, and apoptosis. Also, highly enriched pathways related to the immune system and signal transduction were shared between both tissues. These findings highlight a novel approach to evaluating candidate antigens for sea lice vaccine development, improving the antigens screening in the SHK-1 cell line model.

## 1. Introduction

One of the most expensive branches of aquaculture is salmon farming, mainly due to the high costs to combat viruses, bacteria, and parasites that threaten fish health and welfare. Several chemical agents and drugs have been used to control fish diseases, which has caused pathogen resistance to these treatments, in addition to safety problems [1,2]. Due to the cost-effectiveness of vaccines, they have been gaining attention as a prophylactic method to prevent viral and infectious salmon diseases [3]. Atlantic salmon is the primary salmon species in farming worldwide [4], and many vaccine formulations have been proposed to control infectious diseases that affect it. For instance, vaccines have been developed for *Vibrio anguillarum* [5], *Yersinia ruckeri* [6], salmonid alphavirus (SAV) [7], *Moritella viscosa* [8], infectious pancreatic necrosis virus (IPNV), infectious salmon anemia virus (ISAV) [9], *Lepeoptheirus salmonis* [10,11], and *C. rogercresseyi* [12,13,14]. Most candidate vaccines are developed with an attenuated or killed microorganism component or a particular surface protein or toxin. Usually, fish vaccine production pathways involve the use of a specific antigen that stimulates the fish’s innate and adaptive immune response [1,3]. Thus, vaccines formulated with antigens produced by recombinant DNA technology are increasingly used in the aquaculture industry [3,15]. However, commercial validation is an expensive and lengthy process. Currently, one of the biggest challenges is finding an antigenic molecule that enhances a protective immune response related to pathogen antigens. Furthermore, the in vivo platform for selecting candidate antigens is very complex.

In recent years, fish cell culture has emerged as a promising in vitro model for toxicity, immunological studies, and disease diagnostics [16]. The particularities of fish cell lines, such as ease of maintenance, culture conditions, and the feasible reproducibility of the research, make this a perfect model for in vitro evaluation. Therefore, it is crucial to carefully choose cell lines to represent cellular functions similar to those of a living organism [17]. Several fish immune response studies have used fish cell lines derived from macrophages and leukocytes. Furthermore, studies have used fish cell lines to evaluate the immune response against antigens and establish a correlation with in vivo vaccine trials [16]. For instance, recombinant proteins and synthetic peptides from rhabdoviral hemorrhagic septicemia virus (VHSV) were evaluated as vaccines in suspensions of trout blood leukocytes and lymphoid organs cultures, showing an increase in leukocyte proliferation and recognition by anti-VHSV Mab. Thus, it was suggested to use cell lines as an antigen prospection method [18,19,20]. For instance, a study performed in the SHK-1 cell line exposed to *M. viscosa* antigens showed an expression modulation of pro-inflammatory genes [21]. In addition, in vitro cell assays have been employed to predict the effect of possible control treatments for pathogens. For instance, macrophage-enriched cell cultures from Atlantic salmon head kidney stimulated with IgM-beads were infected with the intracellular pathogen *Piscirickettsia salmonis.* This treatment evaluated whether macrophage cultures reversed the lysosomal evasion mechanism of *P. salmonis* during the infection process. Notably, the results showed an increase in the macrophages’ lysosomal activity and a reduction in pathogen viability, indicating the effectiveness of this approach in developing therapeutic strategies against *P. salmonis.* [22].

The sea louse *C. rogercresseyi* is an ectoparasite with a high economic impact on the Chilean salmon industry [23]. Delousing drugs control this ectoparasite; however, they often have a low efficacy due to loss of pesticide sensitivity in lice populations [24,25]. Thus, the vaccine developed for sea lice control is a sustainable alternative for the salmon industry. The ectoparasites vaccine development was primarily initiated due to the reduced efficacy of the pesticide to tick control in the bovine sector. One of the most successful ectoparasite vaccines has been the recombinant vaccine against the cattle tick *Boophilus microplus* [26]. The antigen’s search for ectoparasite vaccines has focused on proteinases/peptidases and their inhibitors secreted for the ectoparasite [27]. Thus, understanding parasite infestation’s molecular mechanisms is essential to developing and designing new control strategies based on biotechnological tools. Increasing studies in the genomics of numerous species have explored and clarified the biological process and molecular functions involved in the host-parasite interactions [28]. In this sense, the *C. rogercresseyi* genome assembly [29] allows for the identification of new vaccine candidates against the sea lice that affects the salmon industry. Among the different parasite proteins involved in the infestation, the secretory/excretory proteins (SEPs) play an essential role, changing the cell host environment by suppressing their immune system [30]. Thus, the characterization of SEPs involved in the sea louse infestation process contribute to the identification of antigens for vaccine design. Pathogen proteases are a significant secreted compound during the infection, facilitating the parasite feeding processes, homeostasis, and avoiding the host’s immune response modulation [30]. One of the main proteases in the parasite’s secretome is cathepsin [31], which has been associated with protein degradation in lysosomes, nuclei, and cytosol [32,33]. In *C. rogercresseyi,* 56 cathepsin-like proteins have been identified, expressed in all sea louse developmental stages [34]. Recently, our research group used the reverse vaccinology pipeline to identify the cathepsin and Perithopin proteins, characterized in the early developmental stages of *C. rogercresseyi* and formulated as vaccine prototypes for sea lice control. The transcriptomic response of Atlantic salmons immunized with the prototypes was mainly related to the modulation of metabolic processes, energy production, and metal ion handling, which has been essential for the host response to the early *C. rogercresseyi* stages. Furthermore, the group vaccinated with the cathepsin antigen showed a significant parasite burden reduction, making it a promising vaccine candidate [12,35]. Despite these results, the sea lice vaccine development bottleneck is in vivo testing, because it is necessary to use a large amount of fish to evaluate several recombinant protein candidates, dosage, and polyvalent formulation strategies. Thus, using cell lines to assess novel antigens could speed up the production process of recombinant vaccines and simplify the study of their impact on the transcriptional immune response. This study explored a cell-based platform as a tool for antigen prospection to vaccine evaluation and future development. Thus, the transcriptomic profile of two study models, the SHK-1 cell line and Atlantic salmon head kidney (HK), was compared in response to the Cr-cathepsin exposition.

## 2. Materials and Methods

### 2.1. Expression of Cr-Cathepsin Protein in E. coli

For the present study, a cathepsin-encoding mRNA sequence previously identified and characterized by our work group was selected as a recombinant test antigen [12]. Briefly, the nucleotide sequence was chemically synthesized with codon optimization based on *E. coli* codon preferences, and was subsequently cloned into a pET30a expression vector (GenScript, Piscataway, NJ, USA). The plasmid construct obtained, pET30a-Cr-cathepsin, was transformed into the *E. coli* BL21 (DE3) competent cells. Isolated colonies were collected and grown overnight at 37 °C at 250 rpm in Luria Bertani medium (LB) supplemented with 50 µg/mL kanamycin. A culture of 500 mL was inoculated with 5 mL of cells cultured and grown at 37 °C until an OD_600_ between 0.5 and 1 was reached. The recombinant protein expression was induced with 1 mM isopropyl-β-D-thiogalactoside (IPTG) (Thermo Fisher Scientific, Waltham, MA, USA) for 5 h. Bacterial cells were harvested by centrifugation at 3500× *g* for 20 min at 4 °C, and washed once with 20 mM Tris-HCl, pH 7. The pellet was resuspended in a lysis/solubilization buffer (20 mM Tris-HCl, 10 mM Imidazole, 300 mM NaCl, 6 M Urea, pH 8) and a protease inhibitor tablet (Thermo Scientific, Waltham, MA, USA). The cell disruption was performed by sonication (Qsonica sonicators, Newtown, CT, USA) for 10 min (5 s on, 10 s off) at 95% amplitude. The lysed cells were centrifuged at 3500× *g* for 20 min at 4 °C, and the supernatant was used in the following purification steps. 

### 2.2. Purification of Cr-Cathepsin Protein by Immobilized Metal ion Affinity Chromatography (IMAC)

The supernatant was filtered by a syringe filter (0.45 μm) and was loaded on a His Trap FF 5 mL affinity column (GE Healthcare, Chicago, IL, USA). Purification was carried out on the AKTA Prime Plus system (GE Healthcare). The column was equilibrated with 20 mM Tris-HCl, 10 mM Imidazole, 300 mM NaCl, with a pH of 8 and a flow of 1 mL/min y 0.5 MPa. Subsequently, the protein was eluted using an Imidazole gradient started with 50 mM. The Cr-cathepsin protein was eluted between 200–250 mM of imidazole. The elution fractions were dialyzed against PBS for complete refolding. The protein concentration in the final samples was determined using the BCA kit (Thermo Fisher Scientific, Waltham, MA, USA). Proteins were identified by 12% SDS-PAGE analysis under reducing conditions and the western blotting of anti-His-HRP (Thermo Fisher Scientific, Waltham, MA, USA) (Appendix A).

### 2.3. SHK-1 Cell Viability Assay

Cell viability was quantified in the SHK-1 cell line (ECACC 97111106) exposed to Cr-cathepsin using alamarBlue™ cell viability reagents (Thermo Fisher Scientific, Waltham, MA, USA). AlamarBlue™ was used to monitor the cell proliferation and the cytotoxicity of agents [36]. The protocol was adjusted for fish adherent cells exposed to the extracellular agent. SHK-1 cells were seeded at 1 × 10^6^ cells/well on a 96-well black plate and incubated overnight at 18 °C in L-15 GlutaMax, 10% FBS. The medium was aspirated, and the cells were treated with five different concentrations of recombinant Cr-cathepsin protein diluted in L-15 GlutaMax, 10% FBS. Two-fold dilutions of Cr-cathepsin were used, starting from 200, to 12, 5 ng/mL. The same treatment was employed for the control group exposed to Bovine Serum Albumin (BSA). Furthermore, untreated cells and a control well with no cells were included in the assay. After 24 h of incubation at 18 °C, the medium was replaced with 100μL of alamarBlue™ solution in L-15 GlutaMax, and the plates were incubated for one hour at 18 °C, according to the manufacturer’s instructions. The quantification of fluorescence was carried out in a Synergy H1 Hybrid reader microplate reader (Agilent Technologies Inc., Santa Clara, CA, USA) at 560 nm excitation and 590 emissions. The average of fluorescence (arbitrary units) with the corresponding standard deviation was calculated from four replicates for each condition evaluated. The viability was plotted as the difference in percentage reduction in emitted fluorescence between the treated and control cells. This test allowed for the selection of the minimum concentration at which a decrease in cell viability was observed, which was defined as the working concentration for the subsequent assay of SHK-1 cells exposed to recombinant Cr-cathepsin.

### 2.4. SHK-1 Cell Line Stimulation with Recombinant Cr-Cathepsin Protein

The SHK-1 cell line was grown at 18 °C in T75 cm^2^ flasks (Thermo Fisher Scientific, Waltham, MA, USA) in L-15 GlutaMax medium (Gibco, Thermo Fisher Scientific) supplemented with 10% Fetal Bovine Serum (FBS) (Gibco, Thermo Fisher Scientific). When SHK-1 cells were confluent, they were plated at 1 × 10^6^ cells/well in a 6-well plate and incubated overnight in L-15 GlutaMax, 10% FBS. The medium was aspirated and changed to a new medium with 100 ng/mL of recombinant Cr-cathepsin protein diluted L-15 GlutaMax, 10% FBS. The Cr-cathepsin treatment was performed for 24 h in triplicate. The control group was SHK-1 cells in medium L-15 GlutaMax, 10% FBS, and 100 μg/mL of Bovine Serum Albumin (BSA). Lipopolysaccharide (LPS) solution (Thermo Fisher Scientific) was used as experimental stimulation control. After treatment, the morphological cell change was evaluated and registered using the EVOS^TM^ M5000 Imaging System microscope (Thermo Fisher Scientific, Waltham, MA, USA).

### 2.5. In Vivo Evaluation of Recombinant Cr-Cathepsin as an Immunogen

The vaccine prototype was formulated at 100 µL per dose with 30 µg of Cr-cathepsin, in a ratio of 30% Cr-cathepsin/70% adjuvant, using the commercial adjuvant Montanide^TM^ ISA 761 VG (Seppic, Paris, France). In addition, a control vaccine with PBS and adjuvant was formulated. *Salmo salar* of 100 gr were acclimatized for two weeks in the experimental laboratory of the Marine Biological Station, University of Concepción, Dichato, Chile. Fish were injected intraperitoneally and divided into two experimental groups, each one with 20 fish per tank, considering three replicates per experimental group. Samples of head kidney (HK) tissue were taken before immunization and 7 days post-immunization. Samples were preserved in RNAlater^®^ RNA Stabilization Reagent (Ambion^®^, Life Technologies^TM^, Carlsbad, CA, USA) and stored at -80 °C until subsequent RNA extraction.

The animal protocol developed in this research was approved by the Ethics, Bioethics and Biosafety Committee, University of Concepción, Chile. This research was carried out following the recommendations of the International Guiding Principles for Biomedical Research Involving Animals (Council for International Organization of Medical Science and The International Council for Laboratory Animal Science, 2012).

### 2.6. Transcriptome Profiling of SHK-1 Cell Line and Salmon HK Exposed to Recombinant Cr-Cathepsin Protein

The experimental cell groups were stimulated with recombinant Cr-cathepsin, and the control group (BSA) was prepared for transcriptome profiling. Three plates from each experimental cell group were harvested and separately pooled through 0.05% Trypsin-EDTA (Gibco, MD, USA). The total RNA was then isolated using TRizol Reagent (Ambion^®^, Life Technologies^TM^, Carlsbad, CA, USA), following the manufacturer’s instructions. At the same time, the total RNA of salmon HK was isolated from each experimental fish group using Trizol Reagent, following the manufacturer’s instructions. The isolated RNA was evaluated by the TapeStation 2200 (Agilent Technologies Inc., Santa Clara, CA, USA) using the R6K Re-agent Kit. RNA samples with RIN > 8.0 were used for library preparation. Subsequently, double-stranded cDNA libraries were constructed using the TruSeq RNA Sample Preparation Kit v2 (Illumina^®^, San Diego, CA, USA). Three biological replicates were sequenced by the Hiseq (Illumina^®^, San Diego, CA USA) platform in Macrogen Inc. 

RNA sequencing reads of all samples were de novo assembled using the CLC Genomics Workbench v21 software (Qiagen Bioinformatics, Aarhus, Denmark). The assembly was performed using the following settings: mismatch cost = 2, insert and deletion costs = 3, contig length > 200 bp, similarity = 0.9, length fraction = 0.8, with automatic bubble and word sizes. RNA-Seq analyses were conducted in the same software to calculate the gene expression of each dataset using the same settings for costs and similarity fraction as for the assembly. Transcript per million (TPM) values were considered the unit for gene expression analyses. Statistical comparisons among TPM values by the experimental group were obtained by calculating the fold change against the control group using a multi-factorial statistic based on a negative binomial GLM implemented in CLC Genomics software. Contigs with fold change values > |4| and FDR *p*-value < 0.01 were considered as differentially expressed and were extracted for gene annotation. Differentially expressed contigs (in any group against the control) were blasted against the SwissProt database [26] by BlastX considering expect value = 10, word size = 11, match/mismatch = 2/−3, and gap costs = 5 (existence)/2 (extension). All of the sequences with an E-value < 1 × 10^−6^ were considered as correctly identified in the protein database.

### 2.7. Gene Ontology and Pathways Enrichment Analysis

For the enrichment analyses, exclusive and common differentially expressed contigs in SHK-1 cells and salmon HK were used. Differentially expressed contigs were annotated by Gene Ontology criteria (GO) into Biological Processes (BP) and Molecular Functions (MF) hierarchies using the Blast2GO plugin in the CLC Genomics software and default parameters. The KAAS—KEGG Automatic Annotation Server [37] was used for ortholog assignment and pathway mapping using the genome of *S. salar* as a reference.

### 2.8. Long Non-Coding RNAs Expression Analyses

The lncRNAs analysis was performed using the Atlantic salmon lncRNAs database reported by Valenzuela- Muñoz et al. [38]. An RNA-seq analysis for lncRNA was conducted with the same protocol described above for mRNA using CLC Genomics Workbench v21 software (Qiagen Bioinformatics, USA) using the SHK-1 cells and the salmon HK data set. Correlation analyses were performed using TPM values of shared top 55 up- and downregulated lncRNAs (fold change ≥ |4|) and selected coding transcripts related to immune response, iron homeostasis, and transcription factors, among others. Pearson’s correlation was conducted using the Corrplot package v.0.92 [39] written in R language, considering all of the samples simultaneously (*p*-value < 0.01 and *r*-value > |0.9|).

## 3. Results

### 3.1. Cr-Cathepsin Cytotoxicity Evaluation

First, the recombinant protein Cr-cathepsin associated with the secretome of the sea louse [12] was expressed and purified (Appendix A). The cell viability on the SHK-1 cell line exposed to recombinant Cr-cathepsin was determined. After 24 h of exposure, the protein concentrations of 25 and 50 ng/mL did not show morphological differences between the treated and the control cells; however, at 100 ng/mL of Cr-cathepsin, vacuole formation was observed (Appendix A). Furthermore, cell viability was not affected by the different recombinant protein concentrations testing (Appendix A), demonstrating that the antigen did not cause cytotoxicity damage. This assay determined that 100 ng/mL was an adequate work concentration for stimulating the cell line. Here, we propose a cell-based antigens prospection workflow for selecting candidate vaccines in Atlantic salmon against *C. rogercresseyi* (Appendix A).

### 3.2. Comparative Transcriptome Profiling between SHK-1 Cells and Salmon Head Kidney Tissue (HK) in Response to Recombinant Cr-Cathepsin

The de novo assembly encompassed the raw data of all samples sequencing; a total of 142,551 contigs were generated with an N50 equal to 458 pb and an average length of 323 pb. For easier data analysis and interpretation, the two control groups (SHK-1 exposed BSA and salmon HK exposed PBS) were assembled as a pool and renamed as the control group (Ctrls). An RNA-seq analysis evaluated the global transcription profiles in SHK-1 cells and salmon HK exposed to Cr-cathepsin. Interestingly, the transcriptome modulation was strongly clustered due to the study model, showing four differential clusters of upregulated transcripts (Figure 1A). Cluster 1 contigs of SHK-1 cells exposed to Cr-cathepsin were highly expressed compared with the other groups. Among the annotated genes we highlight the *nuclear factor of activated T-cells cytoplasmic 3-like isoform X2*, *Cell wall protein IFF6-like*, *SPARC*, *Heat shock factor-binding protein 1*, *cathepsin H*. Meanwhile, cluster 2 was associated with upregulated contigs in salmon HK, presenting annotated genes such as *Hemoglobin subunit α-4 and β*, *Cytochrome c oxidase polypeptide*, *Metalloendopeptidase*, and *Ferritin*. Furthermore, the hierarchical analysis showed a cluster 3 in the control group, with highly expressed contigs independent of treatment and tissue. Finally, in cluster 4, the contigs were up-regulated in SHK-1 cells and salmon HK exposed to the antigen. Among the annotated genes most expressed in cluster 4 were *ORF2 protein*, *Tripartite motif-containing protein 25*, *TBC1 domain family member 15-like*, *MHC class I-related gene protein-like isoform X1*, and *protein NLRC3-like*. Nevertheless, the expression level of common contigs in cluster 4 also showed a differentiation dependent on tissue (Figure 1A). The annotation and TPM values of the four most upregulated contigs clusters are listed in Appendix A.

Differential expression genes (DEGs) analyses showed a clear differentiation among SHK-1 and salmon HK exposed to Cr-cathepsin compared to the control group. A total of 17,825 and 16,563 contigs were exclusively expressed in SHK-1 cells and salmon HK exposed to Cr-cathepsin, respectively. Notably, both tissues shared 10,949 contigs, representing 24.15% of the total transcriptome response (Figure 1B). From exclusive DEGs annotation, of the top 50 up and downregulated transcripts (fold change ≥|4| and *p*-values < 0.05), identified genes such as *Fibrillin-2*, *Transcription factor HIVEP3-like isoform X2*, *Mucin-19-like*, and *protein NLRC3-like* were upregulated exclusively in SHK-1 cells exposed to Cr-cathepsin; otherwise, the *Coronin*, *Plastin-2*, *Interleukin-6 receptor subunit α*, and *MHC class II antigen β chain* was founded in the most downregulated genes in SHK-1 cells exposed to Cr-cathepsin (Figure 1C). Furthermore, *ATP-binding cassette sub-family A member 1-like isoform X2*, *complement C1q-like protein 2*, and *cell surface A33 antigen-like isoform X2* were exclusively upregulated in salmon HK exposed to Cr-cathepsin. Meanwhile, *Ras-related protein Rab-27B*, *protein NLRC3-like isoform X3*, and *cathepsin K* were downregulated in salmon HK exposed to Cr-cathepsin (Figure 1D) (Appendix A).

### 3.3. Function Enrichment of Exclusive and Common DEGs mRNA

To assign potential functions of exclusives and common DEGs from SHK-1 cells and salmon HK, GO and KEGG enrichment analyses were performed. The GO analysis revealed that several Molecular Functions (MF) associated with binding were shared for exclusive and common DEGs, such as ion binding, metal ion binding, cation binding, and catalytic activity was similarly enriched in terms of interception counts (Figure 2A). Furthermore, hydrolase activity was enriched in the exclusive DEGs of both tissues. In addition, ATP binding, anion binding, adenyl ribonucleotide binding, and adenyl nucleotide binding were only enriched on common DEGs. On the other hand, the enrichment of Biological Process (BP) terms has a different behavior (Figure 2B). Several BP were significantly enriched in the exclusive DEGs of SHK-1 cells, and the same processes were found to be enriched in the common DEGs, such as the regulation of the biological process: signaling, signal transduction, cell communication, response to stimulus, and biological regulation. Instead, macromolecule modification was highly enriched in the exclusive DEGs of salmon HK and in the common DEGs. Moreover, exclusive DEGs of salmon HK triggered BPs that were only enriched in this group, including the organonitrogen compound metabolic process, protein metabolic process, proteolysis, and the developmental process (Figure 2B).

The KEGG enrichment of exclusive and common DEGs showed variations in the number of genes that were involved in each pathway (Figure 3). Several pathways involved in signal transduction were highly enriched, such as the PI3K-Akt signaling pathway, MAPK signaling pathway, Rap1 signaling pathway, and the JAK-STAT signaling pathway. Notably, pathways related to immune response, such as the C-type lectin receptor signaling pathway, NOD-like receptor signaling pathway, Chemokine signaling pathway, T cell receptor signaling pathway, Th1 and Th2 cell differentiation, melanogenesis, and the toll-like receptor signaling pathway were mainly enriched in exclusive and common DEGs. Moreover, apoptosis, spliceosome, and cytokine-cytokine interactions, among others, were enriched (Appendix A). It is worthy of note that the exclusive DEGs for SHK-1 cells showed a higher enrichment of the KEGG pathways; nevertheless, the fact that each pathway is similarly enriched demonstrates the possibility of a shared transcriptomic response.

### 3.4. RNA-seq Relevant Immune Genes

RNA-seq was conducted using 30 transcripts selected from common DEGs to explore the transcriptional response of immune-related genes in response to Cr-cathepsin (Figure 4). The analysis showed three main clusters, where the genes *TLR13*, *TLR5*, *Il-15R*, *IL-13R*, *MHC class I*, *TGF β* were actively transcribed in SHK-1 cells exposed to Cr-cathepsin. Meanwhile, protein *NLRC3*, *IL-3R*, *IL-1R*, *IFN type I*, *TLR2*, *IL-2R*, and *IL-18R* clustering highly transcriptional modulation in salmon HK exposed to Cr-cathepsin. The transcriptome profiling evinced a gene highly expressed in both study-models: Interferon-induced GTP-binding protein Mx3, and indicated the downregulation of other genes, such as *Heat shock protein 70 kDa* and *Haptaglobin*, compared to the Ctrls group. Overall, the hierarchical heatmap evinced a clusterization of transcriptional response based on tissue. Congruently, *TLR6*, *Ferritin*, and *Cathelicidin* were actively transcribed in SHK-1 cells and the control group, while *MHC class I antigen*, *IL-17R*, *TNF alfa*, *Mx3 protein*, and *Haptaglobin* were highly co-expressed in the salmon HK and the control group (Figure 4).

### 3.5. LncRNA Identification and Expression

From 3763 Atlantic salmon lncRNA previously reported by Valenzuela-Muñoz et al. [38], a total of 3485 were found into the SHK-1 cell line and salmon HK exposed to recombinant Cr-cathepsin. A heatmap of the RNA-seq analysis showed a different transcriptional pattern for each group analyzed (Figure 5A). Here, two hierarchical clusters of upregulated lncRNA were clearly differentiated according to tissue and were separated according to the expression pattern of the control group (clusters 1 and 2). Meanwhile, the lncRNAs upregulated simultaneously in SHK-1 cells and salmon HK were grouped in cluster 3. From the DEGs analysis between the Cr-cathepsin exposed groups and the controls, a total of 1194 and 1337 lncRNA were differentially expressed exclusively in SHK-1 cells and salmon HK exposed to Cr-cathepsin, respectively. Meanwhile, 954 lncRNA were identified as common lncRNA sequences (Figure 5B). Both tissues showed a similar fold change variation in upregulated lncRNAs. However, in the SHK-1 cells exposed to Cr-cathepsin, a significant lncRNA downregulation was observed compared to those in salmon HK (Figure 5C).

### 3.6. Expression Correlation between Shared mRNA and lncRNA

A correlation matrix by Pearson analysis was conducted between the top 55 up and downregulated lncRNAs (fold change ≥ |4|) expressed in SHK-1 cells and salmon HK; and the shared transcripts that scored for relevant genes related to immune response, iron homeostasis, transcription factors, and apoptosis, among others. Co-expression analyses demonstrated that two main clusters of lncRNA have a high and low correlation with coding transcripts (Figure 6). The lncRNAs of cluster 1 showed a positive correlation mainly with various interleukins receptors (*IL-1R*, *IL-2R*, *IL-17R*, *IL-18R*), toll-like receptors (*TLR2*, *TLR6*), and B and T cell receptors. The cytosolic regulator of innate immunity, protein NLRC3-like, was also highlighted for its strong positive correlation with the lncRNA of cluster 1. On the other hand, the lncRNAs grouped into cluster 2 showed a highly positive correlation with another group related to cytokines (*IL-3R*, *IL-13R*, *IL-15R*, *TNF α-induced protein 8*, *IFN-induced GTP binding protein Mx3-like*) and toll-like receptors (*TLR5*, *TLR13*). Notably, several cathepsins and genes related to apoptosis (*Caspase 6-like*, *Caspase 14-like*, and *apoptosis facilitator Bcl2-like*) had a positive correlation with the lncRNA set. *Cathelicidin* and *Heat shock 70 kDa protein 12A-like* exhibited the highest positive correlation lncRNAs grouped into cluster 2 (SsLnc_0005382, SsLnc_0135576, SsLnc_0043072, SsLnc_0061656, SsLnc_0100586, and SsLnc_0064622). 

## 4. Discussion

Developing successful vaccines for commercial application against sea lice in salmon farming production is time-consuming and expensive. Several potential vaccine candidates against sea lice have been reported [10,11,12,13,14,40,41,42]. Hence, parasite transcriptomic studies have provided valuable information for understanding the molecular mechanisms involved in the host-parasite interaction [43,44]. Our research group recently reported transcriptional and morphological changes in lice exposed to immunized fish with the IPath^®^ vaccine [13,14]. The findings revealed that IPath^®^ immunization significantly changed the molecular response of sea lice against their host. Moreover, in Atlantic salmon, the effect of a recombinant chimeric protein against *L. salmonis* was evaluated, showing an efficiency of 56% and the enhancement of systemic and local immunity during the host-parasite interaction [10,45]. The identification of protective antigens is a fundamental step in the development of effective vaccine candidates against ectoparasites. Parasite secretory/excretory products play an essential role in tissue penetration, digestion, shedding, and host immune response evasion [27]. Among the parasite SEPs, cathepsins proteins are widely found [46]. Several studies have taken advantage of cathepsins’ functional relevance and antigenic potential for use as vaccines, such as in mammals and chickens [47,48]. In *C. rogercresseyi’s* life cycle, a different expression profile of cathepsin genes has been reported [34]. For instance, a previous study reported a high expression level of several cathepsin *B*, *D*, *F*, *K*, *L*, *S*, and *Z*-like genes in the copepodid stage, which were previously associated with evasion of the host’s immune system, molting, or feeding [34]. Previously, our research group reported using a recombinant *C. rogercresseyi* cathepsin as a vaccine for sea lice control. In the study, Atlantic salmon injected with the recombinant cathepsin showed a sea lice burden reduction of 57% [12]. However, one of the biggest challenges for developing and designing fish vaccines is that the in vivo evaluation is time-consuming [1]. This study evaluated the use of a cell-based platform for sea lice antigens prospection, comparing the transcriptome variation of two study models: SHK-1 cells and salmon HK tissue exposed to the recombinant Cr-cathepsin. 

The SHK-1 cell line was derived from the leucocytes of Atlantic salmon head kidney [49] and has macrophage and dendritic cell characteristics. The main difference between the cell line concerning macrophages is the inability to kill bacteria or pathogens [49]. However, it has excellent potential for immune response studies. The salmonid head kidney is a hematopoietic organ and contains many T and B lymphocytes, macrophages, and melano-macrophages [50,51,52]. Interestingly, the SHK-1 cell line can synthesize melanin [53], and melanogenesis pathway genes have shown a high expression level in salmon head kidney and SHK-1 cells [51,53].

The statistical comparisons showed differences in the transcriptomic profile between SHK-1 cells and salmon HK, with 24.15% of the transcripts being shared between the two study models, which could be primarily attributed to antigen response. The GO classification in common DEGs showed a molecular functions enrichment associated with metal ion binding and catalytic activity. Ions play a crucial role in different physiological aspects, specifically in the immune system cells, where they must maintain strict homeostasis [54,55]. Ion channels and transporters regulate ion concentrations inside T and B cells, and thus could be modulating the functions of these cells [56,57]. Furthermore, metal ions have been associated with innate immune system activation through different mechanisms. Among them, it has been described that they can directly activate pathogen recognition patterns. In addition, they are involved in the inflammasome activation and, through the release of alarmins, enhance necrotic cellular death [58]. Interestingly, an adjuvant role has been attributed to specific metal ions that promote dendritic cell migration and antigen presentation to T cells-metal specifics [59].

Previously, in Atlantic salmon infected with *C. rogercresseyi*, iron transport genes such as *Hepcidin*, *Transferrin receptor*, and *Haptoglobin* have been reported to be upregulated in the head kidney during sea lice infestation, suggesting a nutritional immune response [60]. Iron regulation plays a fundamental role in fish immunity, protecting host tissues from oxidative stress and limiting iron availability to pathogens [61]. Here, essential genes of iron regulation were upregulated, and *Ferritin H*, and *Ferritin M* was highly activated in SHK-1 cells exposed to Cr-cathepsin and the *Haptoglobin* gene in salmon HK. Furthermore, the *Transferrin receptor* was upregulated in both study models, suggesting that the cells present iron homeostasis mechanisms. Moreover, pro- and anti-inflammatory cytokines and acute-phase proteins can regulate the transcription of genes involved in iron transport, directly contributing to iron homeostasis [62]. These results support the theory that SHK-1 cells possess the mechanisms to generate nutritional immunity against *C. rogercresseyi* cathepsin.

Congruently, the pathways involved in the immune response are widely enriched in the common DEGs, suggesting that they play a relevant role in the immune response against the recombinant antigen. Cytokine-cytokine receptor interactions, the TNF signaling pathway, the NF-kappaB signaling pathway, and the NOD-like receptor signaling pathway were highly enriched in common DEGs. Different authors suggest that the antigen recognition by immune cells in the head kidney triggered cytokine–cytokine receptor interactions, and then NF-kappaB and TNF signaling pathways enhancing several transcriptions of immune genes and inflammatory factors [63,64]. Furthermore, the activation of Nod-like receptor (NLR) members of the PRR family has been associated with cellular stress, and thus induces the inflammasome assembly. In addition, the endogenous stimuli trigger the activation of signaling pathways such as NF-kappaB and mitogen-activated protein kinases (MAPKs), which induce a proinflammatory response [65,66]. Moreover, the NLRs play an essential role in the fish innate system, recognizing a lipopolysaccharide and polyinosinic-polycytidylic acid (poly I:C) from bacterial pathogens [67]. In this study, the *NLRC3*-like protein was upregulated in salmon HK, showing a strong response of Atlantic salmon’s innate immune system against a *C. rogercresseyi* cathepsin protein.

Although the signaling pathways and immune-related pathways are enriched in the exclusive and common DEGs, this does not mean that the expression of genes involved in the pathways is the same. Interestingly, the RNA-seq analysis of selected shared immuno-genes revealed different transcriptional expression patterns. Several Interleukin receptors were upregulated on the salmon HK exposed to Cr-cathepsin, such as *IL-1R*, *IL-2R*, *IL-3R*, *IL-17R*, and *IL-18R*; meanwhile, the transcriptions of *IL-13R* and *IL-15R* were highly activated in SHK-1 cells. The expression of pro-inflammatory cytokines has been reported in Atlantic salmon infected with *L. salmonis*, observing the up-regulation of *IL-1R type 1* in the head kidney [68]. Notably, the recombinant protein Cr-cathepsin induces an *IL-1R* down-regulation in SHK-1 cells. Previously, the effect of three fractions of *L. salmonis* SEPs purified by size-exclusion chromatography has been reported in adherent head kidney leucocytes [69]. The effects of *L. salmonis* SEPs in SHK-1 cells showed an *IL-1β* inhibition after stimulation. *IL-1β* regulates several genes, including *IL-8*, in the CXC family of chemokines. These molecules recruit specific subsets of leukocytes, leading to infection and inflammation, playing a fundamental role in host defense [70]. This study showed a significantly enriched chemokine signaling pathway in both study models, suggesting a host cytokine induction in response to the sea louse recombinant Cr-cathepsin protein.

Furthermore, the toll-like receptor signaling pathway was highly enriched in the KEGG analysis. Interestingly, in head kidney tissue, the *TLR2* was upregulated. Meanwhile, *TLR5*, *TLR6*, and *TLR13* were highly expressed in SHK-1 cells in response to Cr-cathepsin. The *TLR13* gene has been described as the most abundant Toll-like receptor type in the head kidney of Atlantic salmon and Coho salmon [71]. The overexpression of *TLR13* observed in Coho salmon infested with *C. rogercresseyi* was associated with fish resistance to ectoparasite infestation [71]. In our current study, the high expression of *TLR13* in SHK-1 cells could be used as molecular markers for antigen evaluation for a vaccine developed for sea lice control.

Advances in high-throughput sequencing have allowed for a deepening in transcriptomic studies, revealing many transcripts that until now do not have a recognized potential coding. The long non-coding RNAs have been considered as potential regulators of different biological processes [38]. Recently, in teleost fish such as Atlantic salmon, Coho salmon, and rainbow trout, it has been found that 2.1% of total transcriptome response against sea lice infestation corresponded with lncRNA sequences [38]. In this study, a total of 3485 putative long non-coding RNAs were identified in the SHK-1 cell line and in salmon HK exposed to recombinant Cr-cathepsin. The differential expression analysis identified 1194 and 1337 exclusive lncRNA in SHK-1 cells and salmon HK, respectively. Meanwhile, 954 lncRNA were identified as common lncRNA sequences. Interestingly, as with the hierarchical clustering of coding transcripts, the lncRNA showed tissue-dependent transcription patterns. The co-expression analysis revealed the potential regulation of lncRNAs over the expression of coding genes involved in relevant biological processes. Remarkably, the sea lice infestation in Atlantic salmon has been described close to the location among lncRNAs and coding genes related to cell migration and transcriptional regulation [72]. Furthermore, several studies have demonstrated the modulation of innate and adaptive response pathways by lncRNA [73], and during the bacterial and viral infection in rainbow trout and Atlantic salmon [74,75]. Furthermore, during the *C. rogercresseyi* infestation of the same species, lncRNA enhanced the modulation of genes associated with the regulation of the immune response, development, cell proliferation, and stress response [38]. This study emphasizes the strong regulation of the lncRNAs of Atlantic salmon over genes related to immune response, iron homeostasis, inflammatory response, and apoptosis, among others. Moreover, it highlights the correlation between *cathepsin* genes and a high number of lncRNAs that are differently modulated. 

## 5. Conclusions

Significant changes in the transcriptome profiling were found in SHK-1 cells and salmon HK after Cr-cathepsin protein exposition compared with the controls. However, the SHK-1 cells and salmon HK exposed to recombinant Cr-cathepsin shared a transcriptomic response associated with the immune system. The signal transduction pathways and system immune-related pathways were significantly enriched in the common DEGs mRNA of SHK-1 cells and salmon HK. Moreover, the lncRNAs expression analysis reveals a model-specific transcription pattern; and it indicates that they are highly correlated with several transcripts involved in the immune response, iron homeostasis, the inflammatory response, and apoptosis. The current study demonstrates that it is possible to use the culture of cell lines for the initial screening of antigens to develop sea lice vaccines, reducing the time consumption associated with the antigen’s selection. However, in the following step, an in vivo assay is required to validate the candidate vaccines and determine their efficacy in sea lice reduction.

## Figures and Tables

**Figure 1 genes-14-00905-f001:**
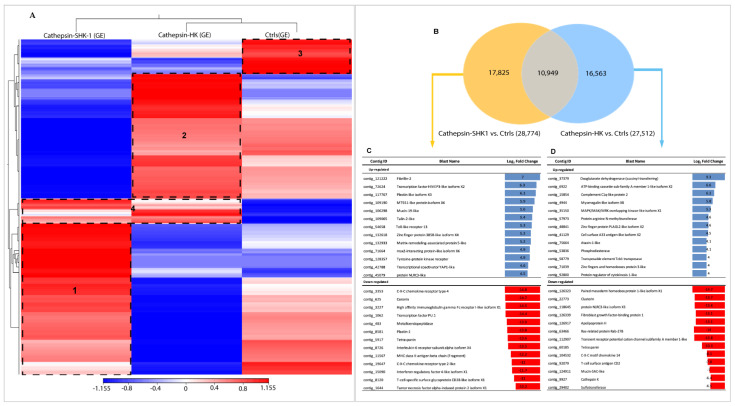
Transcriptome profiling of mRNA from the SHK-1 cell line and salmon head kidney (HK) exposed to recombinant Cr-cathepsin. (**A**) Heatmaps for each treatment were constructed with the TPM (transcripts per million of reads) values of mRNAs and grouped by hierarchical clustering based on the Manhattan distance with average linkages. A red color indicates up-regulated mRNAs, and blue represents down-regulated transcripts. (**B**) Top-20 DEGs expressed in SHK-1 vs. control. (**C**) Venn diagrams of DEGs among tissues (SHK-1 cell line and HK exposed to Cr-cathepsin) vs. control group (|fold change| > 4, *p*-value < 0.05). Up/down-regulated transcripts were calculated from the comparisons between the SHK-1 cell line vs. control (17,825 transcripts), HK vs. control (16,563 transcripts), and shared (10,949 transcripts). (**D**) Top-20 DEGs in HK vs. control.

**Figure 2 genes-14-00905-f002:**
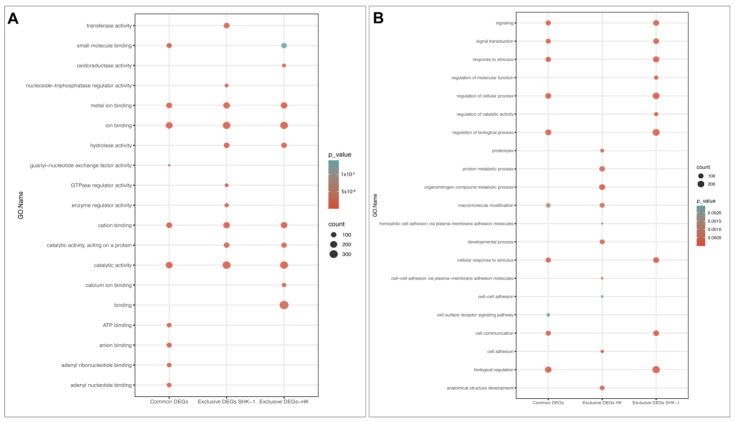
GO enrichment analysis for exclusive and common DEGs of the SHK-1 cell line and salmon HK exposed to recombinant Cr-cathepsin. (**A**) Top-20 of molecular function. (**B**) Top-20 of biological process.

**Figure 3 genes-14-00905-f003:**
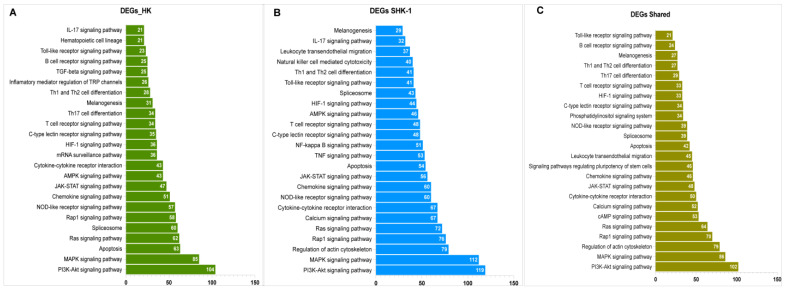
KEGGs enrichment analysis for exclusive and common DEGs mRNA of SHK-1 cell line and salmon HK exposed to recombinant Cr-cathepsin. (**A**) Top pathways of exclusive DEGs mRNA of salmon HK. (**B**) Top pathways of exclusive DEGs mRNA of exclusive of SHK-1 cell line. (**C**) Top pathways of common DEGs mRNA salmon HK and SHK-1 cell line.

**Figure 4 genes-14-00905-f004:**
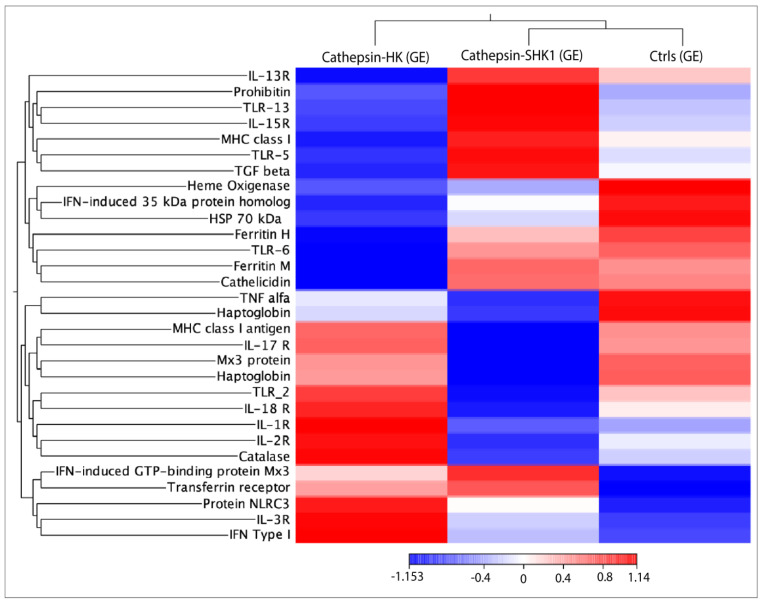
RNA-seq analysis of 30 immune-related genes selected from common DEGs in response to Cr-cathepsin. Heatmaps for each treatment were constructed with the TPM (transcripts per million of reads) values of mRNAs and grouped by hierarchical clustering based on Manhattan distance with average linkages. A red color indicates up-regulated mRNAs, and blue represents down-regulated transcripts.

**Figure 5 genes-14-00905-f005:**
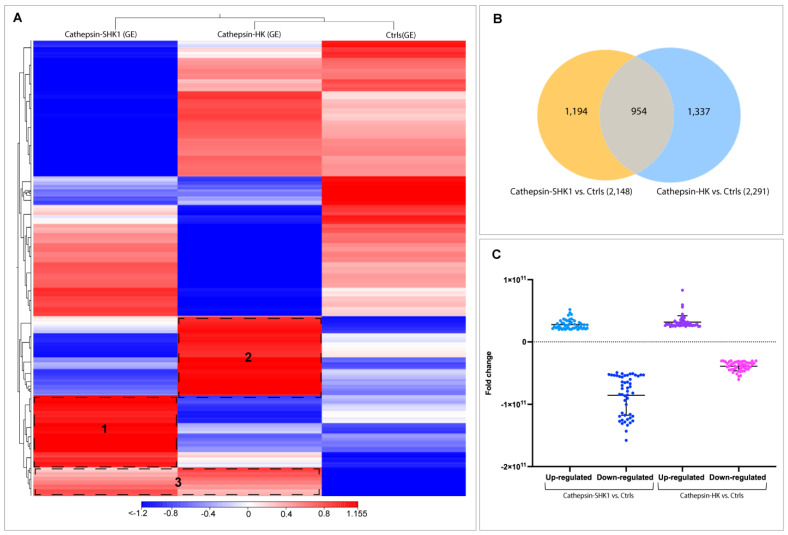
Transcriptome profiling of lncRNA from the SHK-1 cell line and salmon HK exposed to recombinant Cr-cathepsin. (**A**) Heatmaps for each treatment were constructed with the TPM (transcripts per million of reads) values of lncRNAs and grouped by hierarchical clustering based on Manhattan distance with average linkages. A red color indicates up-regulated lncRNAs, and blue represents down-regulated transcripts. (**B**) A Venn diagram representing the significantly expressed lncRNA among tissues (SHK-1 cell line and salmon HK exposed to Cr-cathepsin) vs. control group (|fold change| > 4, *p*-value < 0.05). (**C**) LncRNA fold-change variation up/down-regulated in SHK-1 cell line and salmon HK exposed to Cr-cathepsin vs. the control group.

**Figure 6 genes-14-00905-f006:**
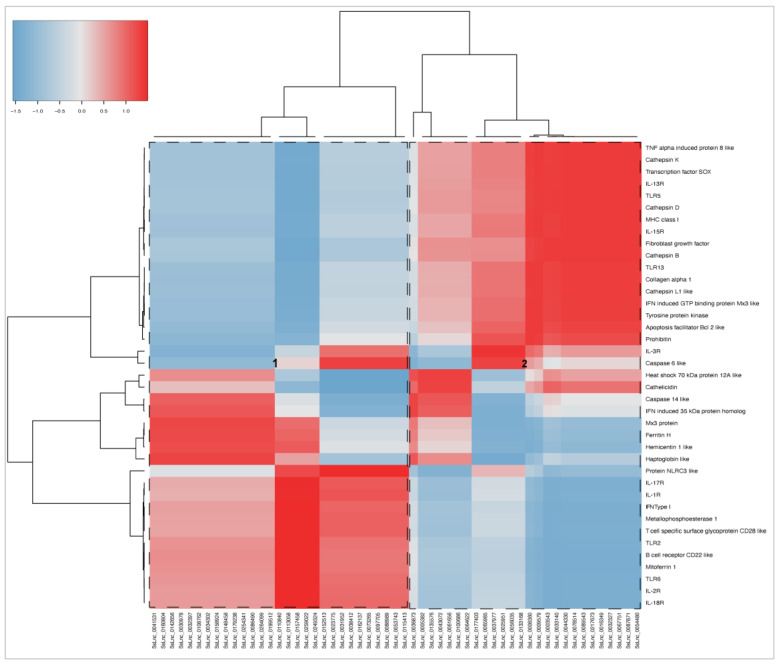
Correlation analyses among the top 55 upregulated shared lncRNA and selected coding transcripts related with immune response, iron homeostasis, and transcription factors. The correlation matrix was based on Pearson’s correlation calculation using transcript fold change values. Positive correlations of expression levels are represented in red, and negative correlations are represented in blue.

## Data Availability

BioProject ID (PRJNA954714).

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
