# Peer review of "Comparative Transcriptomics in Atlantic Salmon Head Kidney and SHK-1 Cell Line Exposed to the Sea Louse Cr-Cathepsin"

_genes, 2023, doi:10.3390/genes14040905_

Round 1

Reviewer 1 Report

Correction of a few typing error errors, and grammatical and stylistic deficiencies - which I have marked on the attached manuscript - may further enhance the quality of the article.

Reviewer 2 Report

The authors, Leal et al.,  have addressed a serious problem for aquaculture that is in need of improved solutions to fight parasitic infections. The approach is scientifically sound and the work provides some interesting results for following studies to advance this area of fish health. There are some issues with the use of English. Below I have highlighted some examples that need correction. The Discussion section is also in need of serious revision. Throughout there are errors that need to be corrected before publication (ejs. : Blas2Go should be BLAST2Go; evidenced [Line 305] should be “evinced”). Pay attention to the use of articles, verb tenses and plurals, but also general semantics. Some of the writing is unclear for me as a native speaker and for non-native speakers in the journal audience it will likely be even more ambiguous.

On a more general note, there should be some space in the Discussion section given to the lack of agreement between the SHK cells and the head kidney results. Whenever a cell line is analyzed for effects by a stimulant it is an observation that is out of context with the rest of the animals cells and the feedback loops that exist in an intact organism. Some feed back is negative and negates the positive stimuli, or vice versa. If there are differentially regulated genes in the whole animal that do not show up in the cell line they could still be valid targets for investigation.

Line 44: it has been gaining space to - (???)

Line 46: For instance, has been developed - (what has been developed)

Line 51: that stimulates the innate and adaptive immune fish response - (fish innate …)

Line 52: technology are increasingly in the aquaculture industry - (increasingly used?)

Line 61: studies have used in vitro methods in fish cell lines to - (used fish cell lines …)

Line 65: These results highlighted the using of primary methods to test candidate´s antigens before in vivo assay. (???)

Line 67: demonstrating the effectiveness of the model to evaluate immune response. -(poor grammar)

Line 73: pathogen viability reduction, (reduction in pathogen viability)

Line 76: However, it has been observed low efficacy of pharmacological methods due - (???)

Line 82: Thus, understanding the parasite molecular mechanisms displayed for a successful infestation – (???)

Line 94: 56 cathepsin-like has been identified,  - (Cathepsin-like proteins?)

Line 101: for sea lice vaccine development is the in vivo testing of several candidates' recombinant proteins, dosage, and polyvalent formulation strategies. (???)

Line 221: Otherwise, the cell assay alamarBlue™ allowed determining that exposure to – (poor grammar)

Line 464: Please revise as: The current study demonstrates that it is possible to use cell culture for screening candidate vaccines that can be effective against sea lice.
